# T cell response to SARS-CoV-2 infection in humans: A systematic review

Madhumita Shrotri[1,2‡], May C. I. van Schalkwyk[3‡], Nathan Post[1], Danielle Eddy[2], Catherine Huntley[1], David Leeman [2], Samuel Rigby[1], Sarah V. Williams[1], William H. Bermingham [4], Paul Kellam[5], John Maher[6,7], Adrian M. Shields[8], Gayatri Amirthalingam[2], Sharon J. Peacock[2,9], Sharif A. Ismail [2,10,11]*

**1** Faculty of Public Health and Policy, London School of Hygiene and Tropical Medicine, London, United Kingdom, **2** National Infection Service, Public Health England, London, United Kingdom, **3** Department of Public Health, Environments and Society, London School of Hygiene and Tropical Medicine, London, United Kingdom, **4** Department of Clinical Immunology, University Hospitals Birmingham, Birmingham, United Kingdom, **5** Department of Infectious Disease, Faculty of Medicine, Imperial College London, London, United Kingdom, **6** School of Cancer and Pharmaceutical Studies, King's College London, London, United Kingdom, **7** Department of Immunology, Eastbourne Hospital, Eastbourne, United Kingdom, **8** Clinical Immunology Service, Institute of Immunology and Immunotherapy, University of Birmingham, Birmingham, United Kingdom, **9** Department of Medicine, University of Cambridge, Cambridge, United Kingdom, **10** Department of Primary Care and Public Health, Imperial College London, London, United Kingdom, **11** Department of Global Health and Development, London School of Hygiene and Tropical Medicine, London, United Kingdom

‡ These authors are joint first authors on this work.
* sharif.ismail15@imperial.ac.uk

**Data Availability Statement:** All relevant data are within the manuscript and its Supporting Information files.

**Funding:** The authors received no specific funding for this work. MCIvS is funded by a NIHR Doctoral

## Abstract

### Background

Understanding the T cell response to SARS-CoV-2 is critical to vaccine development, epidemiological surveillance and disease control strategies. This systematic review critically evaluates and synthesises the relevant peer-reviewed and pre-print literature published from 01/01/2020-26/06/2020.

### Methods

For this systematic review, keyword-structured literature searches were carried out in MEDLINE, Embase and COVID-19 Primer. Papers were independently screened by two researchers, with arbitration of disagreements by a third researcher. Data were independently extracted into a pre-designed Excel template and studies critically appraised using a modified version of the MetaQAT tool, with resolution of disagreements by consensus. Findings were narratively synthesised.

### Results

61 articles were included. 55 (90%) studies used observational designs, 50 (82%) involved hospitalised patients with higher acuity illness, and the majority had important limitations. Symptomatic adult COVID-19 cases consistently show peripheral T cell lymphopenia, which positively correlates with increased disease severity, duration of RNA positivity, and non-survival; while asymptomatic and paediatric cases display preserved counts. People

Fellowship (Ref NIHR300156). JM acknowledges the support of the National Institute for Health Research (NIHR) Biomedical Research Centre based at Guy's and St Thomas' NHS Foundation Trust and King's College London. SAI is supported by a Wellcome Trust Clinical Research Training Fellowship (Ref No 215654/Z/19/Z). The views expressed in this paper are those of the authors only, and do not necessarily represent those of the NHS, the NIHR, PHE or the Department of Health.

**Competing interests:** All authors have read the journal's policy and declare: no support from any organisation for the submitted work; JM is chief scientific officer, shareholder and scientific founder of Leucid Bio, a spinout company focused on development of cellular therapeutic agents; no other relationships or activities that could appear to have influenced the submitted work. This does not alter our adherence to PLOS ONE policies on sharing data and materials.

with severe or critical disease generally develop more robust, virus-specific T cell responses. T cell memory and effector function has been demonstrated against multiple viral epitopes, and, cross-reactive T cell responses have been demonstrated in unexposed and uninfected adults, but the significance for protection and susceptibility, respectively, remains unclear.

## Conclusion

A complex pattern of T cell response to SARS-CoV-2 infection has been demonstrated, but inferences regarding population level immunity are hampered by significant methodological limitations and heterogeneity between studies, as well as a striking lack of research in asymptomatic or pauci-symptomatic individuals. In contrast to antibody responses, population-level surveillance of the T cell response is unlikely to be feasible in the near term. Focused evaluation in specific sub-groups, including vaccine recipients, should be prioritised.

## Introduction

Severe Acute Respiratory Syndrome Coronavirus 2 (SARS-CoV-2), the novel pathogen causing coronavirus disease 2019 (COVID-19), has spread globally and was declared a pandemic by the World Health Organization (WHO) on 11th March 2020 [1]. At the time of writing, there have been around 57.9m confirmed cases and 1.4m deaths reported to the WHO [2]. Lack of pre-existing immunity to this novel and highly infectious betacoronavirus is likely to be responsible for the extraordinary surge in cases worldwide.

There has been an unparalleled global effort to characterise the immune response to SARS-CoV-2 infection, and to develop and test vaccine candidates at unprecedented speed. Understanding the patterns in individual- and population-level immunity will be key to informing future decisions on implementation of non-pharmacological interventions, broader public health policies, and strategies for vaccine delivery. While there is a rapidly growing body of literature on the antibody response to SARS-CoV-2, much less has been published on the T cell response, despite its critical importance in antiviral immunity and vaccine development.

There are principally three areas of interest; firstly, the role of T cells in viral control and immunopathogenesis during acute SARS-CoV-2 infection; secondly the role of T cells in establishing durable protective immunity against reinfection; and finally, the relevance of pre-existing cross-reactive cellular immunity from endemic human coronaviruses (HCoV), or SARS-CoV-1 [3].

This paper focuses on summarising current understanding of the cellular response to SARS-CoV-2 infection, specifically exploring the role that T cell-mediated immunity might play in resistance to severe infection, clinical and virological recovery, and long-term protection–while recognising the dynamic interdependence of the two arms of the adaptive immune response. It is the second of two linked papers summarising results from a wide-ranging systematic review of peer-reviewed and pre-print literature on the human adaptive immune response to SARS-CoV-2 infection [4].

## Methods

A systematic review was carried out according to the Preferred Reporting Items for Systematic Reviews and Meta-Analyses (PRISMA) guidelines. The protocol was pre-registered with PROSPERO (CRD42020192528).

### Patient and public involvement

There was no patient or public involvement in the conceptualisation or design of this review.

### Identification of studies

Keyword-structured searches were performed in MEDLINE, Embase, COVID-19 Primer and the Public Health England library [5] for articles published between 01/01/2020-26/06/2020. A sample search strategy can be found in S1 Appendix in S1 File. We also consulted subject area experts to identify relevant papers not captured through the database searches.

### Definitions, inclusion, and exclusion criteria

We included studies in all human and animal populations, and carried out in all settings (laboratory, community and clinical—encompassing primary, secondary and tertiary care centres), relevant to our research questions. We excluded case reports, commentaries, correspondence pieces or letter responses, consensus statements or guidelines, and study protocols. We included studies reporting on any aspect of the T cell response irrespective of follow-up duration, and on correlates of that response. We defined "correlates" to include (among others) age; gender; ethnicity; the presence of intercurrent or co-morbid disease e.g. diabetes, cardiovascular, chronic respiratory disease; and primary illness severity, proxied by the WHO's distinction between "mild", "moderate", "severe" and "critical" COVID-19 [6], or by requirement for intensive care.

### Selection of studies

Studies were independently screened on title, abstract and full text by two team members (working across four pairs), and disagreements arbitrated by one of the review leads.

### Data extraction, assessment of study quality, and data synthesis

Data were extracted in duplicate from each included study into a bespoke Microsoft Excel template (S2 Fig in S1 File). Where both pre-print and peer-reviewed versions of a report were returned through searches, results were extracted from both if substantial differences in reported data were identified; if little difference was found, only the peer-reviewed version was retained.

Critical appraisal for each included study was performed in duplicate using a version of the MetaQAT 1.0 tool that was adapted for improved applicability to the basic science and laboratory-based studies that are common in this field [7]. The adapted MetaQAT tool was used to gather both qualitative feedback on study quality and scaled responses (yes/no/unclear) to questions around study reliability, internal and external validity, and applicability, with narrative assessment of quality used to inform review findings. Full details of this process can be found in S3 Appendix and S4 Fig in S1 File.

Due to the degree of methodological heterogeneity across included studies, formal meta-analysis was not performed. Results are synthesised narratively in the sections that follow.

### Ethical approval

This was a systematic review based on analysis of openly published secondary data and did not involve humans. No ethical approval was required.

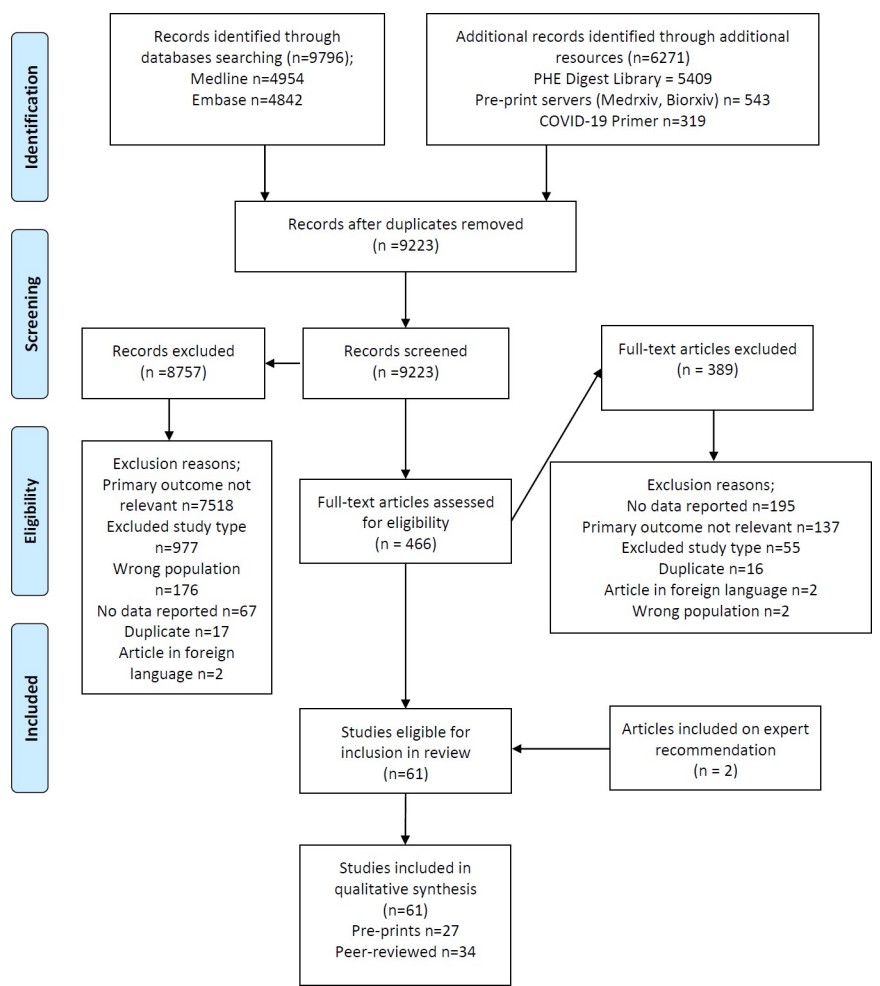

**Fig 1. PRISMA flowchart documenting the search and screening process for this review.**

## Results

### Descriptive overview of included studies

A total of 9,223 records were identified through searches conducted for the review after de-duplication, and a further five through expert consultation, of which 61 papers were included (see PRISMA flowchart in Fig 1).

Key characteristics of included studies are further summarised in Table 1. Of the included reports, 34 (56%) were peer-reviewed journal papers [3, 8–40]. Two animal-based, basic science studies were included [30, 41] but the overwhelming majority of reports were in humans, for which the most common designs were case-control (n = 26, 43%) [10–12, 14, 17, 21, 23, 25–29, 31, 32, 37, 38, 40, 42–50] and cohort (n = 22, 36%) [8, 13, 15, 16, 20, 22, 34, 36, 39, 51–62]. 50 studies (82%) considered participants sampled from hospital settings [8, 9, 11–29, 31–40, 42, 43, 45, 46, 48–50, 53, 54, 56–65]. Most studies originated from China (n = 32, 52%) [11–13, 15–21, 23–29, 31–39, 43, 45, 50, 53, 61, 63]. Only five studies (8%) specifically examined cellular responses in children [15, 19, 33, 35, 39]; while only one study examined differences by gender [24], and none by ethnicity (see Table 2).

**Table 1. Summary of descriptive statistics for included studies.**

| Characteristic | Number | Percentage of total |
|---|---|---|
| Publication type | | |
| Pre-print | 27 | 44% |
| Peer-reviewed | 34 | 56% |
| Study design | | |
| Case control | 26 | 43% |
| Cohort | 22 | 36% |
| Case series | 7 | 11% |
| Basic science study | 3 | 5% |
| Narrative review | 1 | 2% |
| Systematic review with meta-analysis | 1 | 2% |
| Non-randomised clinical trial | 1 | 2% |
| Study population | | |
| Human | 58 | 95% |
| Animal | 2 | 3% |
| Both | 1 | 2% |
| Country from which study population was drawn | | |
| China | 32 | 52% |
| Europe excl. UK (France, Germany, Italy, Spain, Sweden, Netherlands) | 13 | 21% |
| USA | 3 | 5% |
| UK | 3 | 5% |
| Other countries | 6 | 11% |
| Multiple populations | 2 | 3% |
| N/A (lab or animal based) | 2 | 3% |
| Sampling context | | |
| Hospital | 50 | 82% |
| Mixed hospital and community | 1 | 2% |
| Community | 6 | 10% |
| Laboratory (animal) | 2 | 3% |
| N/A (review) | 2 | 3% |

What follows is a narrative synthesis of the main study findings grouped according to topic area. In each section we highlight main limitations of the included papers, with more detailed summaries of each study, the methods and assays applied, as well as specific limitations further elaborated on in the S1 File. Overall, many important study limitations were identified in all topic areas (individual study critical appraisal details are given in S5 Appendix in S1 File), the details and implications of which are explored in the discussion.

## Acute phase T cell response and association with cytokine release syndrome

**General features of the T cell response in the acute phase.** The majority of included papers commented on general aspects of the T cell response to SARS-CoV-2 infection in the acute phase of illness, though the duration of this period was not explicitly defined. Methodological reporting was of variable quality across included studies: in n = 10 papers (16% of the included set) methods were not clearly described, and for the remainder, approaches to quantification of the T cell response varied. For example, Laing *et al.* partnered a total lymphocyte count from a full blood count and flow cytometry to derive estimates of absolute T cell subset counts based on the gated percentages [54], while other studies used direct quantification of

lymphocyte subsets, such as TruCount™ [58]. A majority of the studies used either recognised or in-house flow cytometry tools.

Higher quality studies consistently found evidence for reduction of total peripheral T cell counts in symptomatic adult patients during the acute phase, often accompanied by increased activation of remaining T cells and evidence of functional 'exhaustion', as defined by expression of the markers PD-1 and Tim-3; however, findings regarding specific subsets were more mixed. Three well-designed cohort studies [22, 32, 54] showed reductions in both CD4+ and CD8+ T cell counts in clinical cohorts ranging in size from 30 to 187 patients, while two found evidence of greater reductions in CD8+ (cytotoxic) than CD4+ (helper) T cells [22, 54]. A cohort study (n = 17 patients) only found evidence of reduction in CD4+ but not CD8+ T cell counts on comparing patients with 'aggravated' (or clinically progressive) with non-aggravated disease [34]. A cohort study of 64 patients from Italy showed that T cell frequencies were maintained in patients with mild and asymptomatic disease [51]. Broadly similar findings emerge from a range of high-quality case-control studies, typically with much larger sample sizes. Three hospital-based case-control studies with sample sizes ranging from 102 to 522 patients found evidence of globally reduced lymphocyte counts (CD3+, CD4+ and CD8+ T cells) in the acute phase [12, 26, 37]. These findings were also reflected in two summary reviews [3, 67]. The first, a medium-quality meta-analysis incorporating data on 5,912 patients across 35 published/pre-print reports, showed that total numbers of B cells, T cells and natural killer (NK) cells were all significantly decreased in COVID-19 patients' peripheral blood [67]. This picture of peripheral T cell lymphopenia in COVID-19 patients is reinforced by findings from a larger body of observational studies though many of these had significant methodological limitations [e.g. 16–18]. Notably, four studies considering T cell responses in paediatric COVID-19 cases universally demonstrated comparable T cell counts to healthy paediatric controls, or higher counts when compared against adult cases [19, 33, 35]. The one study to evaluate responses in asymptomatic adult cases (n = 20) found little change in the circulating T cell counts within this group also [51].

Five studies provided more detailed analysis of T cell phenotypes in severe and/or critical disease, with overall suggestions of higher T cell activation with increasing disease severity, alongside depletion of specific subsets that reverses with clinical recovery [13, 27, 51, 60, 62]. A well-conducted study by Anft *et al.* (n = 53) found significant peripheral depletion in critical patients of activated (e.g. HLA-DR+) memory/effector T cells that co-express tissue migratory markers (e.g. CD11a), when compared to severe and moderate cohorts [62]. Lower frequencies of terminally differentiated T-cell subsets (TEMRA) were found in patients with both severe and critical disease. Importantly, recovery from acute respiratory distress syndrome (ARDS) was accompanied by a restoration of CD11a+ T cell subsets. Two studies of critically ill patients identified stronger inflammatory cytokine T cell responses to Spike (S) protein [62], and to S, membrane (M) and nucleocapsid proteins (NP), with greater reactivity by CD4+ compared to CD8+ cells [60] within this group, respectively. Carsetti *et al.* reported an overall increase in activated (e.g. HLA-DR+) CD4+ T cells in 16 patients across both mild and severe disease but found that HLA-DR+ CD8+ cells were specifically increased in severe disease [51]. Two studies also found increased numbers of activated T cells in patients with severe and critical disease, with reversal upon disease remission [13, 27].

Accompanying T cell dysregulation, a cytokine release syndrome (CRS)-like clinical picture occurs in many patients with severe SARS-CoV-2 infection [68]. Elevated levels of many pro-inflammatory cytokines, such as interleukin-6 (IL-6), and to lesser degree, interleukin-10 (IL-10), and tumour necrosis factor alpha (TNF-α) were identified in patients in four studies [3, 69–71]. Concentrations of pro-inflammatory cytokines such as IL-6 positively correlated to severity of disease and with lymphopenia [8, 11, 16, 17, 21, 22, 27, 36, 37, 61, 65, 67]. A large

**Table 2. Evidence on clinical and demographic correlates of T cell response to SARS-CoV-2 infection from studies included in this review (\* disease severity was defined in various ways in included studies; for some according to intensive care unit admission; a number used the Chinese National Health Commission definition [66]).**

| Category | Correlate | Dimension or sub-population | Findings |
|---|---|---|---|
| Clinical | Disease severity\* | Asymptomatic or pauci-symptomatic | 1. • One study evaluated T cell responses in asymptomatic patients (n = 20) and found little change in the circulating T cell frequencies within this group [51]. |
| | | Moderate disease | • Reduced numbers of both CD4$^+$ and CD8$^+$ T cells in moderate and severe cases, alongside increased numbers of activated CD4$^+$ and CD8$^+$ T cells expressing PD-1 or Tim-3; as well as potential reductions in cytotoxic potential and polyfunctionality were reported in one narrative review [3]. |
| | | Severe or critical disease | *Cell counts* |
| | | | • A medium quality meta-analysis found that patients with severe disease had statistically significant, two-fold decreases in both CD4$^+$ and CD8$^+$ T cells, as well as in CD3$^+$ T cells (1.7-fold) and overall lymphocyte number (1.44-fold), alongside statistically significant increases in neutrophils (1.33-fold) and overall leukocytes (1.2-fold) [67]. |
| | | | • A large study (N = 599) reported reduced total, CD4$^+$, and CD8$^+$ T cells being associated with more severe disease, comparing n = 43 ICU-admitted patients with non-ICU-admitted patients, and comparing critical/severe non-ICU patients with mild/moderate non-ICU patients (as per Chinese national definitions\*) [37]. |
| | | | • Other large studies [11, 18, 32, 36] showed comparable findings, and 3 studies also reported reduced CD3$^+$ cells in more severe disease [18, 32, 36]; however, one only found significant cell count differences for critical vs severe disease, and not for severe vs moderate disease [18]. |
| | | | *Cell ratios* |
| | | | • Six studies reported marked increases in CD4/CD8 ratio (due to increases in CD4+ but reductions in CD8+ cells) in severe and critical patients compared to those with moderate disease [22, 25, 36, 62, 65]. The last of these also showed CD8$^+$ T cell counts were much slower to normalise than CD4+ in patients with severe disease [25]. |
| | | | • Two studies however, reported significant reductions in CD4$^+$, but not in CD8$^+$, T cells in severe disease (n = 452), or 'aggravated' disease, defined as clinically progressive at 7 days (n = 17) [26, 34]. |
| | | | • A small study from Iran reported increased CD8 expression in ICU patients relative to healthy controls, quantified by flow cytometry as mean fluorescence intensity (MFI), with no significant differences seen in CD4/CD8 ratio, or CD4+ T cell MFI [40]. |
| | Clinical endpoint | Survival vs non-survival | • Two studies with large cohorts followed up COVID-19 patients until death or discharge, both conducting multivariate analysis. Luo *et al.* (n = 1018), reported significantly lower CD3$^+$, CD4$^+$ and especially CD8$^+$ counts in non-survivors than survivors, and found that CD8$^+$ T cell counts <165 cells/μL (OR 5.93) were independently associated with mortality after adjustment for age, sex and comorbidities [21]. Liu *et al.* (n = 340) reported that lower helper T cells (OR 0.22) and higher CD4/CD8 ratio (OR 4.8) were highly significant predictors of mortality [63]. |
| | | | • Whilst also reporting lower CD8$^+$ counts in non-survivors throughout the disease course, Wang *et al.* (n = 157) also found that non-survivors had lower CD4$^+$ counts only evident in middle and late stages of disease, and that non-survivors had a lower CD4/CD8 ratio [29]. |
| | | | • Based on 28 deaths amongst 187 patients, Xu *et al.* demonstrated that total T cell counts <500/μl, CD3+ counts <200/μl, CD4$^+$ or CD8$^+$ counts <100/μ as well as B cell counts <50/μL, were significantly associated with risk of in-hospital death, however this is only on univariate analysis [32]. |
| | | | • In a cohort of n = 548, Chen *et al.* reported significantly elevated neutrophil-to-lymphocyte ratio (NLR), platelets-to-lymphocytes ratio (PLR), reduced peripheral CD3$^+$, CD4$^+$ and particularly CD8$^+$ counts in non-survivors [36]. He *et al.* (n = 204) reported that T cell levels continued to fall until death in non-survivors, whilst in survivors with severe disease, levels increased after 15 days and normalised after 25 days of treatment [11]. |
| | | RNA persistence | • Four small but high or medium quality clinical cohort studies from China showed that slower resolution of PCR-positivity is associated with reductions in peripheral T cells. |
| | | | • Jiang *et al.* (n = 23) found that the baseline abnormalities in CD3$^+$, CD4$^+$ and CD8$^+$ T cells underwent robust recovery in patients who became RNA negative 2 weeks after diagnosis, whilst they did not do so in those who remained persistently positive [12]. |
| | | | • Liu *et al.* compared 37 cases who remained positive at day 20, with 37 patients at their point of diagnosis, as well as 54 healthy controls, and showed that both the persistently positive and control groups had higher CD3$^+$ and CD4$^+$ levels, suggesting that these subsets do normalise despite viral persistence [45]. |
| | | | • In a similar study, though with a persistence threshold of 15 days, Dong *et al.* (n = 18) also found global reductions across CD3$^+$, CD4$^+$ and CD8$^+$ subsets for persistent positives, which increased between admission and discharge; as well as significant negative correlation between overall T cell count and duration of positive nucleic acid test [38]. |
| | | | • Liu *et al.* (n = 39) also reported higher global T and B cells in patients becoming RT-PCR negative within 14 days [20]. |
| | Co-morbid disease status | | • Three studies considered the effect of comorbid status, all originating from China and spanning patients with non-severe, severe and critical clinical presentations [11, 20, 39]. Two had significant methodological limitations [20, 39]. |
| | | | • One study (n = 204) found significantly lower total lymphocyte and lymphocyte subset counts in patients with comorbidities compared with those without (though "comorbidities" not defined) [11]. |
| | | | • The second (n = 39) found statistically significant differences in CD8$^+$ counts between patients with comorbid disease and those without (p = 0.046), but no difference in CD4$^+$ counts—although here again the range of comorbidities considered was not defined [20]. |
| | | | • The final study compared outcomes in a paediatric cohort with or without "allergic disease" (not clearly defined) and showed no effect on clinical course, total lymphocyte or lymphocyte subset counts [39]. |
| Demographic | Age | Older adults | • A high-quality clinical cohort study and a medium-quality case-control study, both from China, reported lower T cell total and subset counts, including CD3$^+$, CD4$^+$, CD8$^+$ subsets, for older patients aged 60 or over [11, 37]. |
| | | Children | • Four medium-quality studies—1 case control and 3 case series—considered cellular responses in children in samples from China, all showing comparable CD3$^+$, CD4$^+$ and CD8$^+$ counts to healthy paediatric controls, or where the comparison group was adults, higher T cell counts across subsets [19, 33, 35]. However, potential confounders such as disease severity or comorbidities were not controlled for in these studies. |
| | Sex | | • One medium-quality case series (n = 27) from China examined differences in cytokine secretion by sex of cases, showing reductions in CD4$^+$ and CD8$^+$ count for all patients irrespective of gender but more generalised cytokine responses were observed among male participants than females, for IL-6, TNF-α and procalcitonin–although the statistical significance of these differences was not tested [24]. |

peer-reviewed study with 1,018 participants reported over ten-fold increases in IL-6 levels amongst COVID-19 cases, and found that serum IL-6 >20pg/mL was strongly associated with in-hospital mortality (OR 9.78, p<0.001) on multivariable regression analysis [21]. A pre-print

systematic review reported 1.93-fold increases in IL-6 and 1.55-fold increases in IL-10 for severe patients [67]. In line with this, another large study (n = 548) reported significantly increased IL-6 levels in non-survivors compared with survivors [36]. Correspondingly, levels of IL-6 and IL-10 appeared to be negatively correlated with total T cell and subset counts across all included studies, and showed normalisation in tandem with clinical resolution [37]. Findings for other interleukins, IL-1, IL-2, IL-4 and IL-8, were more mixed [11, 16, 17, 27, 37, 61, 67].

**Dynamics of the T cell response over the acute phase.**   Seven studies reported longitudinal data on the T cell response, mostly focusing on within-hospital trends, with a maximum follow-up range of 14–44 days following symptom onset [8, 9, 11, 12, 32, 45, 59]. Overall, the available evidence suggests that peripheral T cell depletion is closely linked with both disease severity and viral load in the acute phase, and recovery of counts can occur rapidly following clinical or virological recovery, especially in more mild disease. Two large and well-conducted case-control studies (n = 103 and n = 187) found that low T cell counts on admission increased steadily over the course of admission. Subsequent recovery of lymphocyte count was roughly consistent with clinical improvement [12, 32]. One study found evidence of significant decreases in counts of CD3$^+$ T, CD4$^+$ T, CD8$^+$ T, and NK cells in COVID-19 patients compared with healthy controls (all p<0.05) on admission. In a subset of n = 23 patients followed up two weeks after initial presentation, those newly negative for SARS-CoV-2 RNA on polymerase chain reaction (PCR) showed the most dramatic recoveries in T cell subset counts [12]. Two studies reported longitudinal trends in detail at regular follow-up intervals; the first, a cohort study from Italy involving 18 patients (nine mild and nine severe cases), found that low total lymphocyte counts in severe cases were stably maintained for up to 20 days post-admission, with little discernible difference between T cell subsets [8]. The second, a French cohort study (n = 15) of predominantly elderly patients admitted to intensive care, found that CD8$^+$ counts fell to their lowest value by days 11–14 after symptom onset (p = 0.03), with recovery thereafter, but noted a slightly later nadir for CD4$^+$ (days 19–23) and with no significant change in the overall CD4/CD8 ratio throughout the 35-day follow-up period [59].

## Correlates of the T cell response

The number of studies addressing demographic and clinical correlates of the T cell response was small and many potentially important variables such as ethnicity were not addressed. Key findings from this literature are summarised in Table 2. The largest single body of work examined relationships between T cell response and disease severity, based predominantly on studies in the hospital setting.

In regard to clinical correlates, peripheral counts appeared undisturbed in asymptomatic disease, significantly depleted in moderate or severe disease, and with disturbances to the CD4/CD8 ratio in severe or critical disease. The single study including asymptomatic cases was of good quality, although limitations included relatively small sample size and poor reporting of sample selection methods [51]. Evidence regarding moderate and severe disease was consistent across several good quality studies with larger sample sizes [32, 37, 62] and was also reflected in two reviews [3, 67].

Lower peripheral T cell counts were associated with non-survival, as reflected in two larger studies which conducted multivariable analyses and found independent associations for specific subsets [21, 63]; with persistent RNA-positivity, primarily in smaller studies with some risk of selection bias; and with older age, including in one large higher quality study [37].

Many studies were limited by poor reporting of sample and control selection methods, and by some variability in their definitions of clinical severity (most as per WHO, however some were based on Chinese national guidance).

## Viral cross-reactivity of T cells

Eight studies explored cross-reactivity of T cells between SARS-CoV-2 and related human coronaviruses within small, adult-only samples of cases and controls [10, 42, 44, 46, 47, 49, 52, 55]. Using activation-induced marker (AIM) assays, Grifoni *et al.* detected SARS-CoV-2-reactive CD4$^+$ T cells against a range of S and non-S epitopes in 12/20 'pre-pandemic' US donors [10] while Weiskopf *et al.* reported low levels of cross-reactivity in only 2/10 'pre-pandemic' German donors [49]. Using an interferon gamma (IFN-γ) enzyme-linked immunosorbent spot (ELISpot) assay, Gallais *et al.* found some T cell cross-reactivity mainly to the S2-domain of the S protein in 5/10 'pre-pandemic' French donors [52] and Le Bert *et al.* found T cells specific to NP and non-structural proteins 7 and 13 (NSP7, NSP13) in SARS-CoV-1/2 unexposed donors [55]. The latter Singapore-based study also reported robust SARS-CoV-2 NP-reactivity in T cells from SARS-CoV-1 convalescents, with these memory cells persisting for 17 years after the SARS outbreak [55].

Amongst controls recruited during the pandemic, but confirmed as antibody- and PCR-negative, S-reactive T cells were demonstrated in 23/68 controls in a high-quality German study [42]; and in 12/14 controls in a smaller Russian study, including one household contact of a COVID-19 case. The latter study also included a smaller group of 'pre-pandemic' donors (n = 10), who had significantly lower frequency and magnitude of reactivity than the controls recruited during the pandemic, hinting at a possible protective effect of cross-reactive T cells [47]. In contrast, Peng *et al.* found no SARS-CoV-2-specific T cell responses in either 'pre-pandemic' or 'during-pandemic' antibody-negative UK controls (n = 19) [46].

Notably, studies consistently found a lower frequency and magnitude of T cell response as well as a differential pattern of immunodominance in reactive unexposed controls relative to SARS-CoV-2 convalescents, with low homology between COVID-19 convalescent T cell epitopes and known epitopes from endemic human coronaviruses (HCoV). An Australian study found that frequencies of T follicular helper (TFH) cells specific to HCoV-HKU1 were higher amongst COVID-19 convalescents (n = 41) than uninfected controls (n = 27), suggesting boosting of HKU1-specific responses following SARS-CoV-2 exposure, and hinting at a coronavirus-specific TFH response (study findings are further elaborated on below in the context of T-cell population characterisation) [44].

The evidence suggests that a degree of cross-reactivity of T cell responses between human coronaviruses may be relatively common; however, the significance of these findings for individual and population susceptibility to SARS-CoV-2 remains unclear. Additionally, the evidence is limited by very small sample sizes, uncertain validity of 'during-pandemic' controls, and heterogeneity in assay methods.

## Characterisation of T cell subpopulations and protective immunity

Twelve studies characterised T-cell subpopulations, including magnitude, functionality and phenotypic characteristics, post-acute COVID-19 infection. Timing of sampling post disease onset and duration of follow-up differed both within and between studies, many of which were conducted on small study populations, with sampling methods rarely reported (S5 Appendix in S1 File). One French contact-tracing study demonstrated SARS-CoV-2-specific T cell responses against structural (S, M, and NP) and accessory proteins in all nine index cases, in samples collected at 47–69 days post symptom-onset, as well as in 6/8 PCR-negative or untested contacts (of whom five were symptomatic), in samples collected up to 80 days post-onset [52]. A UK-based study of COVID-19 convalescents (28 mild cases, 14 severe cases) characterised the T cell response using IFN-γ ELISpot assays on samples taken at least 28 days post symptom onset [46]. A strong and broad SARS-CoV-2-specific T cell response was

generally elicited but varied between individuals. T cell response breadth (p = 0.010) and magnitude (p = 0.002) were significantly higher in patients who recovered from severe disease in comparison to mild cases. Sub-set evaluation demonstrated CD8+ T cells mediated a greater proportion of responses detected to S and M or NP epitopes. No difference in the levels of polyfunctional T cells was observed between mild and severe disease. Differences were observed in the cytokine profiles of CD8+ T cells targeting different viral antigens, with the M/NP-specific CD8+ T cells displaying wider functionality compared to those targeting S-protein (p = 0.0231). In those with mild disease, M/NP-specific CD8+ T cells were significantly higher than S-specific T cells. This trend was not observed in those with severe disease [46].

These findings complement the study by Grifoni *et al.* (discussed above) which found that NP, M and S proteins contain the immunodominant epitopes for both CD4+ and CD8+ T cells [10]. No significant differences in the cytotoxic potential was detected between mild and severe disease. Specific SARS-CoV-2-reactive T cells were not frequently observed in healthy, unexposed individuals. Furthermore, the magnitude of T cell responses in COVID-19 patients correlated with related antibody titres, including anti-S and anti-NP. Another study stimulated peripheral blood mononuclear cells (PBMCs) from 18 COVID-19 patients ranging in disease severity with two overlapping peptide pools spanning the full S region [42]. Twelve patients had detectable CD4+ T cell reactivity against the first peptide pool, which contained N-terminal epitopes including the receptor binding domain (RBD). Fifteen patients displayed reactive CD4+ T cells against the second peptide pool, which contained C-terminal epitopes processing higher homology with HCoVs. Among the non-reactive cases most had critical disease [42].

Le Bert *et al.* assayed peripheral blood T cell responses to NP and NSP7 and NSP13 of the large SARS-CoV-2 proteome using an IFN-γ ELISPOT assay. Samples were obtained from 24 individuals who had experienced mild to severe COVID-19. For all patients, IFN-γ spots were observed following stimulation with NP peptide and nearly all displayed responses against multiple regions of NP. A further sub-analysis demonstrated T cell recognition of multiple regions of SARS-CoV-2 NP among recovered patients (8/9) [55].

Six studies reported on the phenotypic and target profile of T cell subsets. One study performed an in-depth characterisation of humoral and cellular immunity against the S protein in samples taken from 41 adults who had recovered from mild-moderate SARS-CoV-2 infection (five requiring hospitalisation but not mechanical ventilation) and 27 controls. Expanded populations of S-specific memory B cells and circulating cTFH cells (which play a critical role in supporting antigen-specific B cells to initiate and maintain humoral immune responses) were detected [44]. The frequencies of unstimulated cTFH cells were comparable between SARS-CoV-2 convalescent and uninfected groups. In general, robust cTFH cells activity to the SARS-CoV-2 S-protein was observed among the convalescent group, whereas responses to RBD-specific cTFH were significantly lower (p<0.0001). The antigen reactivity of S-specific non-cTFH CD4 memory (CD3+CD4+CD45RA-CXCR5-) cells revealed similar trends with strong recognition of SARS-CoV-2 and smaller frequencies of RBD-specific T cells. High plasma neutralisation activity was also found to be associated with increased S-specific antibody, but notably also with the relative distribution of S-specific cTFH subsets [44].

Another study analysed the T cell response in samples taken from 31 COVID-19 patients [13]. Disease severity was classified in accordance with US National Institute of Health classification system [72], with a total of n = 2, n = 19, and n = 10 participants being categorized as having asymptomatic, mild, and moderate/sever disease, respectively. None of whom required intensive care or oxygen supplementation. A central memory phenotype (CD45RO+, CCR7+), followed by an effector memory phenotype (CD45RO+, CCR7-) were predominate within the S-reactive CD4+ T cell population. An effector memory, followed by the terminal effector cells (CD45RO-, CCR7-) were the predominant phenotypes among antigen-specific CD8+ T cells.

A significant increase in activated (CD38[+], HLA[-]DR[+]) CD4[+] T cells was detected among cases. Further T cell response characterisation showed CD4[+] and CD8[+] T cell activation in response to full-length S-protein exposure, and the M-protein response was significantly stronger (p = 0.0352). A correlation between the magnitude of T-cell and humoral responses was reported (anti-RBD IgG and CD8[+] T-cell response). However, this relationship was weakly statistically significant (r = 0.386 p = 0.0321), whereas an interdependence was reported between the magnitude of CD8[+] and CD4[+] responses (r and p values not presented) [47]. Three additional studies reported on the presence of the effector memory phenotype, two of which studied hospitalised patient populations, and the third study analysed samples from returning travellers. Minervina and colleagues reported detection of T cell clones within both the effector and central memory subpopulations, in samples obtained from two returnees from countries with high SARS-CoV-2 transmission [64]. Similarly, Weiskopf *et al.*, in their study of 10 COVID-19 patients who developed ARDS, reported that peripheral SARS-CoV-2-specific CD4[+] T-cells typically had a central memory phenotype (based on CD45RA and CCR7 expression), whereas the majority of virus-specific CD8[+] T-cells had a CCR7- effector memory (TEM) or TEMRA phenotype [49]. In contrast, a study of four COVID-19 positive paediatric cases with mild disease, and five uninfected controls, found no difference in the effector or central memory phenotypes of the CD8[+] and CD4[+] populations compared with controls [33].

A small study conducted a phenotypic analysis of circulating SARS-CoV-2-specific T cells in samples obtained 20–47 days post positive PCR from individuals recently recovered from mild SARS-CoV-2 infection. The analysis was conducted using combination SARS-CoV-2-specific T cell detection with CyTOF. IFN-γ producing S-specific CD4[+] and CD8[+] T cells were detected, suggestive of a S-specific T helper (Th)1 response, where as Th2 and Th17 lineages were not detected among S-specific CD4[+] T cells [73].

Evidence of potential protective T cell-mediated immunity is provided by one US-based study that measured the cellular response in rhesus monkeys (n = 9 cases, n = 3 controls) upon repeat challenge with pooled S peptides, day 35 post initial infection [30]. Based on IFN-γ ELISpot assays, cellular immune responses were observed in the majority of animals, with a trend toward lower responses in the lower dose groups. Intracellular cytokine staining assays demonstrated induction of both S-specific CD4[+] and CD8[+] T cell responses. Post re-challenge, very limited viral RNA was observed in bronchoalveolar lavage (BAL) on day one following re-challenge in three animals, with no viral RNA detected at subsequent timepoints. In contrast, high levels of viral RNA were observed in the concurrently challenged naive animals. However, these findings to do not exclude the possibility that protection was antibody-dependent rather than due to T cell immunity exclusively, and longer-term analyses are needed [30].

## Discussion

This review narratively synthesises findings from 61 studies examining human T cell responses to SARS-CoV-2 published before the end of June 2020. Given the exceptional speed and volume of developments in COVID-19 research, further evidence has accumulated in the intervening months. In this section we summarise key findings from the review and contextualise them against new data published since our searches were completed in late June 2020; importantly, we have not identified any reports that challenge the central findings of this review.

### Summary of key findings

Acutely, adult COVID-19 patients exhibit a depletion of T cells in the peripheral blood, the extent of which is positively correlated with disease severity, whereas asymptomatic patients

and children tend to have preserved peripheral T cell counts. This suggests an important relationship between pathogenesis and the circulating T cell pool. Observed lymphopaenia in adult COVID-19 patients is likely to be multifactorial in origin, with redistributive effects, apoptotic loss [74], and possibly reduced mobilisation of lymphocytes from bone marrow, all playing a part. Prior work has also shown an association between IL-6 production and blockade of lymphopoiesis; although the extent to which this mechanism operates in COVID-19 has yet to be investigated [75]. Regarding age differences, it has been speculated that children may receive protection from a diverse naive T cell repertoire, with adults of increasing age at higher risk due to immunosenescence [76]. At the time of searching, few studies had explored the relationship between T cells, age and clinical severity, with appropriate statistical adjustment, however, a recent study examining all three branches of the adaptive immune response (CD8 + T cells, CD4+ T cells, and neutralising antibodies), found that older age and scarcity of naïve T cells were associated with un-coordinated adaptive responses and more severe disease [77]. Another recent study reported more robust S-specific T cell responses in adults (mean age 61.05 years) compared with children (mean age 13.34 years) [78].

There is emerging evidence of cytokine over-production, in particular IL-6 and IL-10, as part of immunopathogenesis within COVID-19; however, drivers of these observed changes are still not fully understood. A recent pre-print report from a Brazilian research group describes infection of CD4+ T cells by SARS-CoV-2, with subsequent high expression of IL-10 by infected cells [79]. If these data are robust to peer-review and can be replicated elsewhere, this finding may represent one of the contributory factors. Trials of immunomodulatory agents including those that inhibit the IL-6 pathway in COVID-19 patients are also underway [80], although initial results for one of these agents, tocilizumab, proved disappointing [81].

Although less comprehensive, longer-term data suggest that T cell reductions are transient, with rapid recovery of counts within days to weeks of clinical recovery and PCR negativity. This supports the hypothesis that T cells are sequestered rather than destroyed, although the observation of similarly depleted T cell numbers in the broncho-alveolar lavage samples of severe patients indicates that T cells are not simply recruited *en masse* to infected tissues [82].

In the context of well-recognised variations in COVID-19 clinical outcomes by age, ethnicity and co-morbid status, there is a striking shortage of robust evidence on demographic correlates of the T cell response to SARS-CoV-2. We identified a single study considering gender-related effects on T cells, and eight studies considering cellular responses with age (a majority of these in paediatric patients with or without adult controls). We identified no studies evaluating other potentially important determinants, including ethnicity. These constitute important gaps in the evidence, which persist even in more recent literature, and should be addressed in future studies.

Evidence characterising cellular immune responses suggest enduring T cell immunity, with phenotypic profiles consistent with helper and memory T cell functions and evidence of activity against multiple viral targets. Variation in viral targets is observed between disease severity and based on one study, the breadth and magnitude of the T cell response were significantly higher in patients who recovered from severe compared to mild disease. Responses were also detected in individuals who experienced mild infection. However, this evidence derives from small, observational studies conducted on samples taken from participants at varying time points, and with selection criteria rarely described. The longevity of this T cell immunity and the degree of protection it provides remains unclear, though recent pre-print papers from studies with longer follow-up report durability of virus-specific T cells for as long as 6–8 months following infection [83, 84]. Recent epidemiological and animal model evidence hints at the protective function of T cells [85, 86], and is supported by identification of detectable virus-specific T cell responses in seronegative COVID-19 convalescents [87–89], and in uninfected individuals with known exposure [90].

With regards to T cell cross-reactivity, included studies reported variable prevalence of SARS-CoV-2-reactive T cells in unexposed controls. These studies were limited by small sizes and assay heterogeneity, but there was consensus around the lower frequency and magnitude of T cell responses, and differential epitope dominance, in reactive controls relative to SARS--CoV-2 convalescents. More recent studies conducting detailed characterisation of the T cell epitopes governing cross-reactivity have found similarity with common cold coronaviruses [89, 91], with one study reporting pre-existing T cell responses in 81% of unexposed controls and data suggestive of lower pre-existing cross-reactivity in hospitalised COVID-19 cases compared with mild cases [89]. Several models of the potential impact of pre-existing cross-reactivity on individual and population immunity have been proposed [92], and methodologies allowing distinction between pre-existing T cell responses, and those arising from SARS-CoV-2 infection, are a growing focus of investigation [93].

## Strengths and limitations

This study is the first systematic review on the T cell immune response to SARS-CoV-2, utilising robust methods for searching, screening, and critically appraising both pre-print and peer-reviewed literature. While a number of narrative reviews are available [94, 95], some of which focus on specific aspects of cellular immunity [67, 96], our review is broader in both scope and comprehensiveness, and is intended as a foundation for ongoing systematic evidence synthesis.

Limitations arise from the methodology applied, and from the nature of the underlying evidence. First, while the search strategy was broad in choice of keywords and inclusion of pre-print publications, it is possible that some results were missed, particularly on pre-print servers for which structured searches are more challenging.

Additional limitations arise from the nature of the underlying evidence base on which this review draws. Variations in reporting practice present major challenges for critical appraisal and weighting of evidence. For example, narrative reviews–popular in this field–have limited methods reporting. Further difficulty is introduced through variations in treatment protocols, clinical severity and case definitions used in included studies, and varying methods adopted for T cell counts, functionality, phenotypes, and assay validation. Not only do these factors introduce substantial methodological heterogeneity, thereby limiting quantitative syntheses of data; they are also critical to the study of T cell immunity to SARS-CoV-2 as the assays are evolving and yet to be formally validated and standardised.

Importantly, many of the studies also had significant methodological limitations, most notably, small sample sizes accompanied by minimal reporting on selection methods for participants and controls, which introduces substantial risk of selection bias. This risk is further compounded where only subsets of samples are characterised in greater depth, or small sub-cohorts are followed-up longitudinally, with little explanation of how these sub-groups are selected. Consequently, it is challenging to draw inferences and to generalise findings to the population-level, limiting applications to wider practice and policy. Other issues affecting the validity and reliability of data, such as lack of valid controls and lack of statistical analyses to control for confounders, for example when testing associations with demographic or clinical correlates, are also commonly encountered issues within the evidence base.

Finally, as a consequence of the urgency of conducting research and disseminating findings during this pandemic, academic conventions have often been circumvented. Many findings were initially (and sometimes solely) reported through pre-print papers, which have not undergone the scrutiny of peer-review. Caution should be applied when drawing inferences from these data, and we have taken care in this study to distinguish clearly between preprint and peer-reviewed publications in reporting findings. Furthermore, we noted large variations

in the ethical approval processes that authors of individual studies appeared to have followed, and the extent to which informed consent was sought from participants. The implications for the integrity of future research are potentially grave and will need to be comprehensively addressed in the interests of ethically sound research practice in future.

## Policy implications and onward research questions

Many unanswered questions remain, such as the durability of and protection afforded by virus-specific T cell responses, and their relative importance in protection from reinfection compared with antibodies. More data is also needed on the demographic correlates of T cell responses and the significance of cross-reactive cellular immunity.

An important application of findings from T cell response studies will be towards evaluation of the rapidly growing number of SARS-CoV-2 vaccine candidates, a number of which are now in or emerging from clinical trials [97]. In parallel with clinical data from COVID-19 patients, vaccine developers are frequently reporting on T cell immunogenicity from early phase evaluations. While this is notably lacking for some prominent candidates (including inactivated vaccines from Sinovac [98], Beijing Institute of Biological Products/Sinopharm [99], and the Chinese Academy of Medical Sciences [100]), other frontrunners (including mRNA vaccines by Moderna [101] and Pfizer/BioNTech [102], and non-replicating viral-vectored vaccines by Oxford University [103], Gamaleya Research Institute [104], and CanSino [105]), have successfully demonstrated vaccine-induced T cell responses against S-protein epitopes. While these data are encouraging, given the wide range of potential T cell epitopes, it is worth exploring whether multi-peptide platforms such as traditional inactivated whole-virus, or novel virus-like particles, may provide more robust immunity through harnessing the full potential of the T cell response, as compared with S-focused mRNA and viral-vectored platforms. This is supported by data from recent studies demonstrating that non-S proteins make up the most immunodominant T cell epitopes following infection [106], and that more diverse T cell responses are associated with milder disease [89]. It will also be important to conduct Phase 3 and post-implementation evaluation of vaccine effectiveness in groups with high prevalence of prior infection, such as health and care staff, who will be a priority group for vaccine deployment following licensure. In addition to antibody testing, baseline assessments of virus-specific T cell reactivity are likely to be highly useful for this purpose.

Current estimates of population immunity rely solely on seroprevalence studies, however in the context of evidence for cellular responses in seronegative exposed individuals, and the potential waning of antibody responses over time, current surveillance methods are likely to be underestimating both exposure and immunity. A more developed understanding of the role of T cells in long-term protection will be helpful to policy makers in terms of modelling where population-level immunity lies and informing long-term surveillance and immunisation strategies. However, by contrast with antibody testing–a mainstay of immune surveillance for many communicable diseases–existing T cell assays are difficult to standardise and hard to scale, therefore unlikely to be deliverable at population level within the timeframe of the SARS-CoV-2 pandemic. In the short-term, emphasis may need to be placed on determining the utility of T cell assays to guide clinical and public health actions at the individual level, particularly in patients with immunosuppression, or those at the extremes of age. In parallel, adequately-powered and controlled studies providing deep immunophenotyping of T cells, B cells, and comprehensive characterisation of immune responses in mild or asymptomatic cases, and in vaccine recipients, will yield insights about the interdependence and relative importance of cellular and humoral responses. Over the long-term, development of scalable T cell assays may help to strengthen population immune surveillance systems.

## Conclusions

A complex picture is emerging concerning the T cell response to SARS-CoV-2 infection, including the interplay between compartments of the immune system, and the balance between protective versus pathological responses. Inferences are limited by methodological limitations within studies, and heterogeneity between studies. Evaluation of T cell responses at scale is currently infeasible and the benefits of such an approach as yet unclear. Findings from targeted testing may carry important clinical and policy implications for public health interventions within at-risk sub-groups, for understanding mechanisms of vaccine efficacy, and for informing long-term population immunisation and surveillance strategies.

## Supporting information

**S1 Checklist. Prisma checklist.**
(DOCX)

**S1 File.**
(DOCX)

## Acknowledgments

We thank Professor Mike Ferguson from the School of Life Sciences, University of Dundee, for comments on the research questions and initial outputs from this work; and Professor Mark Petticrew from the Faculty of Public Health and Policy, London School of Hygiene and Tropical Medicine, and Rachel Clark, Head of Evidence & Evaluation in the Research, Translation & Innovation Division at Public Health England, for advice on methodological aspects of this study. We are also grateful to Anh Tran (Senior Knowledge and Evidence Manager), Nicola Pearce-Smith (Senior Information Scientist), Paul Rudd (Knowledge and Evidence Specialist–COVID-19) and James Robinson (Knowledge and Evidence Specialist–North) from Public Health England's Knowledge and Library Services for support in conducting the literature searches on which this review was based.

## Author Contributions

**Conceptualization:** May C. I. van Schalkwyk, Danielle Eddy, Paul Kellam, Gayatri Amirthalingam, Sharon J. Peacock, Sharif A. Ismail.

**Investigation:** Madhumita Shrotri, May C. I. van Schalkwyk, Nathan Post, Danielle Eddy, Catherine Huntley, David Leeman, Samuel Rigby, Sarah V. Williams, Sharif A. Ismail.

**Methodology:** May C. I. van Schalkwyk, Sharif A. Ismail.

**Project administration:** Danielle Eddy, Sharif A. Ismail.

**Supervision:** Gayatri Amirthalingam, Sharon J. Peacock.

**Validation:** William H. Bermingham, Paul Kellam, John Maher, Adrian M. Shields, Gayatri Amirthalingam, Sharon J. Peacock.

**Writing – original draft:** Madhumita Shrotri, May C. I. van Schalkwyk, Nathan Post, Danielle Eddy, Catherine Huntley, Sharif A. Ismail.

**Writing – review & editing:** Madhumita Shrotri, May C. I. van Schalkwyk, Nathan Post, Danielle Eddy, Catherine Huntley, David Leeman, Samuel Rigby, Sarah V. Williams, William H. Bermingham, Paul Kellam, John Maher, Adrian M. Shields, Gayatri Amirthalingam, Sharon J. Peacock, Sharif A. Ismail.

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
