## [Decision Letter · Decision Letter 0]

1 Dec 2020

PONE-D-20-27929

Cellular immune response to SARS-CoV-2 infection in humans: a systematic review

PLOS ONE

Dear Dr. Ismail,

Thank you for submitting your manuscript to PLOS ONE. After careful consideration, we feel that it has merit but does not fully meet PLOS ONE’s publication criteria as it currently stands. Therefore, we invite you to submit a revised version of the manuscript that addresses the points raised during the review process.

We look forward to receiving your revised manuscript.

Kind regards,

Stephen R. Walsh, MDCM

Academic Editor

PLOS ONE

Journal Requirements:

"All authors have read the journal's policy  and declare: no support from any organisation for the submitted work; JM is chief scientific officer, shareholder and scientific founder of Leucid Bio, a spinout company focused on development of cellular therapeutic agents; no other relationships or activities that could appear to have influenced the submitted work."

Reviewers' comments:

Reviewer's Responses to Questions

**Comments to the Author**

1. Is the manuscript technically sound, and do the data support the conclusions?

Reviewer #1: Yes

Reviewer #2: Yes

Reviewer #3: Yes

2. Has the statistical analysis been performed appropriately and rigorously? 

Reviewer #1: Yes

Reviewer #2: N/A

Reviewer #3: N/A

3. Have the authors made all data underlying the findings in their manuscript fully available?

Reviewer #1: Yes

Reviewer #2: Yes

Reviewer #3: Yes

4. Is the manuscript presented in an intelligible fashion and written in standard English?

Reviewer #1: Yes

Reviewer #2: Yes

Reviewer #3: Yes

5. Review Comments to the Author

Reviewer #1: This manuscript attempts to cover systematically cover and review a growing body of information on cellular T cell responses to SARS-CoV-2. They have clearly outlines the process in which they have done this, and discussed the challenges of doing so. One challenge is that because this field is evolving so rapidly, many papers are being published without thorough peer review and this should be discussed a bit more. I would encourage the authors to consider including more of the latest publications post June 2020.

Minor: Spell out RBD for those not familiar with this term. The mention of the Minervina et al., study in line 364-365 seems out of place and does not link up with the rest of the paragraph.

It may be worth mentioning the link of looking at Tfh further upfront e.g. move from line 342-343 to where you first talk about Tfh.

Consider revising to be more consistent for each study sited to include as much information as possible about disease status i.e. mild, severe etc. For example in line 353, its unclear if those are mild cases or severe cases but they just didnt need iC or oxygen supplementation.

The correlation sited in line 361 is weak in my opinion.

Reviewer #2: This systematic review on cellular immune response was well-organized and the narrative was easy to follow which reflects on the capability of the authors to express their message efficiently.

I had issues with several marker names and protocols that need to be spelled in full at first mention.

Reviewer #3: Summary of Research

Shroti M et al have written a systematic review of published literature regarding T cell responses to COVID, using independent keyword-structured literature searches for articles published (56% of final articles selected) or pre-prints available (44% of final articles selected) between the beginning of this year until 26 June, 2020, with a total of 61 articles included in the final analysis. A modified MetaQAT tool was used to critically appraise research articles by 2 independent authors, with a third author as an arbitrator in case of disputes. Overall, the authors conclude there is enough evidence to support that peripheral T cell count inversely correlates with disease severity, and that memory T cell and effector function to multiple viral epitopes are induced by infection. However, most studies had several limitations, highlighting the need for more research in this area.

Overall impression

Strengths included the critical assessment and quality scoring of the articles (as presented in Table 2) as well as duplicate independent analysis of journal articles with tie breaker reviewer.

However, while the authors did provide a good summary of data, there seemed little interpretation/synthesis of results. Often it did not seem to flow well, and seemed more of a disjointed accounting of facts from studies without interpretation or conclusions for each point, or broader digestion/interpretation of results. They state they will explore “the role of T cell mediated immunity in resistance to severe infection, clinical and virological recovery and long-term protection, (lines 92-93)” but discussion in these areas are vague and are stated to be inconclusive.

Major issues:

1. Would benefit from greater synthesis of presented data to provide more generalized conclusions/summary for each section (similar to summary in lines 180-183). For example, lines 204-205 set up the paragraph to discuss 5 more detailed studies, but what is the conclusions that can be drawn from these studies? What is the significance of the depletion of migratory T cells in severe cohorts? In line 222, is there a direct pathway connecting IL-6 and lymphopenia to potentially explain this correlation? For the paragraph starting 236, what do these studies suggest about the dynamics of the T cell response over time during the acute phase or can any overall conclusions be drawn? If not, why not?

2. Would also benefit from greater thought about how the data integrate into a larger picture – for instance, can they draw broad conclusions to form a model of how T cells respond to Covid depending on disease severity?

3. Any comments on how methodology compared between studies? The authors mention that cross-study statistical analysis was not performed due to degree of methodological heterogeneity across studies, but rarely comment on that in the text. They authors mention in their limitations section that many studies had small sample size, which they do point out for most studies, but did not otherwise comment on any particular methodological limitations of studies, which would be useful for a reader to determine weight to place on specific sections and may place the data in a different light.

4. Would benefit from updating.

a. Various reviews on T cell response have come out in recent months (see Toor SM et al, Immunology, Sept 2020, Chen Z and Wherry EJ, Nature reviews immunology, July 2020), although this may be the only systematic review, should consider rephrasing or citing.

b. Last article assessed was prior to July, would be nice to include any significant data from relevant publications since then, even if only in discussion.

c. Vaccine paragraph needs updating given recent interim results (lines paragraph starting line 457), with possible interpretation.

d. Almost half of studies were not peer-reviewed. For those that were pre-prints, recommend reviewing if any have undergone publication since then (similar to what was stated to have been done in lines 135-136).

Minor issues:

1. Overall lack of sufficient reference citation, see lines 172-183; 353-361; 381-390; 396-398; 402-404; 407-408; 415-417, although this is not inclusive of all areas needing improved citation.

2. The MetaQat based analysis of the publications was very effective, but one question, “Does it consider a similar population to the UK” makes it a little less broadly applicable and this reviewer wonders why this was included in the criteria.

3. Would benefit from more discussion of the quality of the data used to make the overall summary statements (such as done in line 425-427) and throughout the paper.

4. Consider renaming article to better reflect the content, specifying T cell responses rather than cellular responses.

5. In text, lines 161, states 34 (58%) were peer-reviewed journals, but in table one, it states 34 (56%). Please correct.

Misc.

1. Supplementary Appendix D is an excellent reference tool for critical analysis of specific studies.

6. PLOS authors have the option to publish the peer review history of their article (what does this mean?). If published, this will include your full peer review and any attached files.

Reviewer #1: **Yes: **One. B. Dintwe

Reviewer #2: No

Reviewer #3: No

---

## [Author Response · Author response to Decision Letter 0]

29 Dec 2020

Department of Primary Care and Public Health

Imperial College London, UK

FAO Stephen R. Walsh, MDCM

Academic Editor

PLoS ONE

30 December 2020

Dear Professor Walsh

Thank you for your email of 01/12/2020 providing reviewer comments on our manuscript “Cellular immune response to SARS-CoV-2 infection in humans: a systematic review”. We have now reviewed these and outline point-by-point responses below. Please note that page or line number references given below refer to the clean version of the manuscript, not the tracked version. 

Editorial comments

Thank you – we have now made amendments to the manuscript accordingly. 

"All authors have read the journal's policy and declare: no support from any organisation for the submitted work; JM is chief scientific officer, shareholder and scientific founder of Leucid Bio, a spinout company focused on development of cellular therapeutic agents; no other relationships or activities that could appear to have influenced the submitted work."

Thank you; we have now made edits accordingly in lines 679-80 of the manuscript. 

Thank you – please see our updated statement below (which now also appears in the manuscript): 

“All authors have read the journal's policy and declare: no support from any organisation for the submitted work; JM is chief scientific officer, shareholder and scientific founder of Leucid Bio, a spinout company focused on development of cellular therapeutic agents; no other relationships or activities that could appear to have influenced the submitted work. This does not alter our adherence to PLOS ONE policies on sharing data and materials.”

Thank you – this has now been moved from the end of the manuscript to the methods section in the main body of the manuscript.

Thank you – we have now updated the manuscript to include the requested details at the end of the manuscript and altered the naming of the supplementary materials per PLOS requirements. 

Reviewer #1

This manuscript attempts to cover systematically cover and review a growing body of information on cellular T cell responses to SARS-CoV-2. They have clearly outlines the process in which they have done this, and discussed the challenges of doing so. One challenge is that because this field is evolving so rapidly, many papers are being published without thorough peer review and this should be discussed a bit more. I would encourage the authors to consider including more of the latest publications post June 2020. 

We agree with the reviewer’s comment regarding the difficulty of keeping pace with the very rapidly evolving literature in this space. To maintain the integrity of the systematic review process, capturing more recent articles in a systematic way would have entailed re-running the searches up to a more recent date but because of the time taken to then run through screening, selection, critical appraisal, extraction and synthesis, we would have faced similar issues of delay. To try to address more recently published material, we have instead included in the discussion reference to literature published up to 14/12/2020 (see the “Summary of Findings” section) drawing on expert input from members of the review team, and also relevant material captured in the Public Health England (PHE) COVID-19 literature digest, an evidence tracking tool produced by the PHE Knowledge and Library Services, and which is publicly available at: https://phelibrary.koha-ptfs.co.uk/coronavirusinformation/#DailyEvidenceDigest. The bulk of this new material can be found under ‘Summary of key findings’ within the Discussion section – but we note that findings from this literature do not changed in substantive ways the findings from the systematic review we conducted. 

Minor: Spell out RBD for those not familiar with this term. The mention of the Minervina et al., study in line 364-365 seems out of place and does not link up with the rest of the paragraph.

Many thanks for these helpful comments. We have given the full term for RBD in line 380. With regard to Minervina et al, we have revised the placement of this particular study and now include a sentence distinguishing between those studies that analysed data from hospitalised patients and the Minerva study which focused exclusively on returning travellers. We hope that this addresses the reviewer’s concerns about the placement of this study.

It may be worth mentioning the link of looking at Tfh further upfront e.g. move from line 342-343 to where you first talk about Tfh.

Many thanks for highlighting that TfH are addressed in two different places within the manuscript. Instead of moving the text from the final section on the characterisation of T-cell populations and protective immunity, we have added a sentence to the preceding section where TfH cells are first introduced, informing the reader that the findings of the study are further elaborated in the following section. We felt that given the importance of the TfH population that the more detailed discussed belonged in the section that focused specifically on T-cell population characterisation.

Consider revising to be more consistent for each study sited to include as much information as possible about disease status i.e. mild, severe etc. For example in line 353, it’s unclear if those are mild cases or severe cases but they just didnt need iC or oxygen supplementation.

We now provide more detail on the disease severity in this study, the classification system used and how many study participants fell within each category.

The correlation sited in line 361 is weak in my opinion.

We have revised this sentence and made it clearer to the reader that the reported correlation was only weakly statistically significant.

Reviewer #2

This systematic review on cellular immune response was well-organized and the narrative was easy to follow which reflects on the capability of the authors to express their message efficiently.

I had issues with several marker names and protocols that need to be spelled in full at first mention. 

We thank the reviewer for raising this. We have now addressed this throughout the manuscript to spell out abbreviations and acronyms at first appearance. 

Reviewer #3

Overall comment: While the authors did provide a good summary of data, there seemed little interpretation/synthesis of results. Often it did not seem to flow well, and seemed more of a disjointed accounting of facts from studies without interpretation or conclusions for each point, or broader digestion/interpretation of results. They state they will explore “the role of T cell mediated immunity in resistance to severe infection, clinical and virological recovery and long-term protection, (lines 92-93))” but discussion in these areas are vague and are stated to be inconclusive.

We thank the reviewer for raising this general issue of interpretation and analysis of results presented in the paper. This review set out to summarise and appraise available evidence at the time at which the searches were conducted – with a view to informing policymakers. A considerable challenge in this analysis has been the diversity of research methods used across research studies, variations in quality, and in study contexts which hamper efforts to integrate findings into a clear overarching narrative. We also contend – in the discussion and elsewhere – that in many of the areas discussed there simply does not yet exist the degree of consensus that would permit clear conclusions to be drawn regarding the role of T-cell mediated immunity in response to infection or in regard to long-term protection. In the discussion, we emphasise that large and important questions – regarding the nature of T cell responses and correlates of immune response among other areas – remain unanswered – indeed this is explicitly recognised at the beginning of the policy implications and onward research questions section. 

In our view, the absence of consensus on critical questions as regards the T cell response is itself a very significant finding – one we highlight in the discussion – that rests on the state of the published literature itself, rather than the quality of the analysis we have undertaken. It is also the key factor motivating the selection of onward research questions we identify in the discussion. 

Major issues:

1. Would benefit from greater synthesis of presented data to provide more generalized conclusions/summary for each section (similar to summary in lines 180-183. For example, lines 204-205 set up the paragraph to discuss 5 more detailed studies, but what is the conclusions that can be drawn from these studies? What is the significance of the depletion of migratory T cells in severe cohorts? In line 222, is there a direct pathway connecting IL-6 and lymphopenia to potentially explain this correlation?

Thank you for flagging this. We have taken the view that the preferred place in the manuscript to provide an overview and synthesis of findings is the discussion – to keep the content of the results section to reporting findings as closely as possible. We have added some clarifying text to try to put the findings on T cell dynamics in more severe disease into context, and commented further on the role of IL-6 under ‘Summary of key findings’ within the Discussion section. One of the considerable challenges in this paper is that the broader clinical implications of a number of the findings we report remain unclear – we hope that work during the current wave may help to answer some of these questions more definitively. 

For the paragraph starting 236, what do these studies suggest about the dynamics of the T cell response over time during the acute phase or can any overall conclusions be drawn? If not, why not? 

Thank you – we have tried to address this question in lines 262-4 of the revised manuscript, but also with expanded commentary in the summary section of the discussion. The implication is that peripheral T cell depletion links closely with disease severity, and that cell counts seem to recover quickly following clinical or virological recovery (especially in milder illness). 

2. Would also benefit from greater thought about how the data integrate into a larger picture – for instance, can they draw broad conclusions to form a model of how T cells respond to Covid depending on disease severity? 

Thank you for this comment. Unfortunately, our overall judgement at this stage is that is not sufficient evidence, nor sufficient agreement between studies that have thus far been published, for such a model of T cell response to be generated according to severity. General models of T cell response to SARS-CoV-2 have been posited in some of the studies we looked at – for instance by Tay and colleagues [46] – but these are aggregate models or visual representations that distinguish only at the level of “healthy” versus “dysfunctional” immune response, with no clear differentiation of the extent of “dysfunction” by symptom severity. 

3. Any comments on how methodology compared between studies? The authors mention that cross-study statistical analysis was not performed due to degree of methodological heterogeneity across studies, but rarely comment on that in the text. They authors mention in their limitations section that many studies had small sample size, which they do point out for most studies, but did not otherwise comment on any particular methodological limitations of studies, which would be useful for a reader to determine weight to place on specific sections and may place the data in a different light. 

We thank the reviewer for this comment. We discuss limitations and methodological comparisons between studies throughout the paper – although within the confines of a systematic review of this size and scope it is not possible to discuss individual studies in full detail, and there are challenges in communicating within a summary narrative the detail of methodological limitations for often highly technical, complex studies. We outline specific methodological limitations to a number of included studies in some detail in the results narrative, and comment in summary terms on the methodological strength of the literature overall in lines 525-66 in the discussion. Finally, supplementary appendix S5 gives key methodological limitations for every individual study included in the review.

We have nevertheless expanded discussion of limitations throughout the manuscript, for example in lines 190-8 and 343-7 but with much greater detail in the discussion under the ‘Strengths and limitations’ section. 

4. Would benefit from updating.

a. Various reviews on T cell response have come out in recent months (see Toor SM et al, Immunology, Sept 2020, Chen Z and Wherry EJ, Nature reviews immunology, July 2020), although this may be the only systematic review, should consider rephrasing or citing. 

Thank you – we have now addressed this in lines 534-9, including by reference to the articles the reviewer lists above.

b. Last article assessed was prior to July, would be nice to include any significant data from relevant publications since then, even if only in discussion. 

Thank you – we have noted above the considerable difficulty of keeping pace with the very rapidly evolving literature in this space. To try to address this we have added material in the discussion to reference more recently published literature up to 14/12/2020 drawing on expert input from members of the review team, and also relevant material captured in the Public Health England (PHE) COVID-19 literature digest, an evidence tracking tool produced by the PHE Knowledge and Library Services, and which is publicly available at: https://phelibrary.koha-ptfs.co.uk/coronavirusinformation/#DailyEvidenceDigest. The bulk of this new material can be found in the “Summary of key findings” within the Discussion section – but we note that findings from this literature do not changed in substantive ways the findings from the systematic review we conducted.

c. Vaccine paragraph needs updating given recent interim results (lines paragraph starting line 457 with possible interpretation.

Thank you – we have addressed this with additional material in lines 585-604.

d. Almost half of studies were not peer-reviewed. For those that were pre-prints, recommend reviewing if any have undergone publication since then (similar to what was stated to have been done in lines 135-136). 

We thank the reviewer for this comment. We have clarified our approach in the section entitled “Data extraction, assessment of study quality, and data synthesis” in the Methods, and discussed the extensive use of preprints in this study as an important limitation under “Strengths and limitations” within the Discussion section. While we are unable to repeat the searches in full, we have updated the review by referencing particularly significant new material published since the end of June within the Discussion section, as outlined in our response above. It would not, however, be possible to update the review selectively, as implied above; the integrity of the systematic review process (and in particular, the efforts to reduce bias) rest on following a clearly documented, and time-defined search process. We followed a protocol pre-published on PROSPERO to help reduce bias and therefore focused our review on reporting publications that met our inclusion criteria and were released during the search period. References we provide to more recently published literature in the discussion are for reference only, to contextualise and complement the main findings, rather than replace them. 

Minor issues:

1. Overall lack of sufficient reference citation, see lines 172-183; 353-361; 381-390; 396-398; 402-404; 407-408; 415-417 although this is not inclusive of all areas needing improved citation. 

We have addressed this by adding further citations where relevant (especially in lines 160-9 at the head of the results section which sets out the span of the papers), but also by moving citations further up in the narrative, as, in some cases, the issues identified relate to single, lengthy paragraphs referring to a single study that was not cited until the final line. 

2. The MetaQat based analysis of the publications was very effective, but one question, “Does it consider a similar population to the UK” makes it a little less broadly applicable and this reviewer wonders why this was included in the criteria. 

We thank the reviewer for this comment. This domain on the MetaQAT tool was included originally because this study was performed as a response to a request for input for policymakers in the UK, and there was a need therefore to situate findings clearly in terms of potentially applicability for/to a UK audience. 

It should also be noted that the applicability element affected principally the quantitative scoring of study quality – our narrative comments on study quality focused on aspects of design, reliability and internal/external validity. As originally used, there were two elements to the MetaQAT: narrative summary of critical appraisal findings using the template (see S4 Figure) and in particular the prompt questions it contains; and a quantitative scoring component for each paper derived from the scaled responses reviewers gave to questions in the tool (which was then used to calculate an aggregate score for each paper, converted into a high/medium/low rating using pre-defined scoring thresholds). Following peer review feedback on the sister paper from the same review (covering the antibody response) – also with PLoS ONE – we have removed the quantitative scoring element in favour of the narrative commentary on study quality, and have also done so for this manuscript to ensure consistency. This is on the basis that it enables more detailed discussion of methodological limitations for each study and avoids the crude clustering encouraged by the high/medium/low rating scheme. 

3. Would benefit from more discussion of the quality of the data used to make the overall summary statements (such as done in line 425-427) and throughout the paper.

As outlined above, we comment in some detail on the quality and methodological limitations of included studies throughout the paper, including in appendix S5. These provide the basis on which summary judgements are made. 

4. Consider renaming article to better reflect the content, specifying T cell responses rather than cellular responses.

Thank you for this comment – we have amended the title as suggested.

5. In text, lines 161, states 34 (58%) were peer-reviewed journals, but in table one, it states 34 (56%). Please correct. 

Thank you – we have now corrected this error and included the correct figures throughout. 

We hope these responses help in addressing the reviewers’ concerns. If you have any further queries please do not hesitate to contact us. 

Yours,

Sharif Ismail

ST4 Public Health Registrar

Wellcome Trust Clinical Research Training Fellow

For and on behalf of the study authors

---

## [Editor Report · Decision Letter 1]

4 Jan 2021

T cell response to SARS-CoV-2 infection in humans: a systematic review

PONE-D-20-27929R1

Dear Dr. Ismail,

We’re pleased to inform you that your manuscript has been judged scientifically suitable for publication and will be formally accepted for publication once it meets all outstanding technical requirements.

Kind regards,

Stephen R. Walsh, MDCM

Academic Editor

PLOS ONE
---

## [Editor Report · Acceptance letter]

12 Jan 2021

PONE-D-20-27929R1 

T cell response to SARS-CoV-2 infection in humans: a systematic review 

Dear Dr. Ismail:

I'm pleased to inform you that your manuscript has been deemed suitable for publication in PLOS ONE. Congratulations! Your manuscript is now with our production department. 

Kind regards, 

on behalf of

Dr. Stephen R. Walsh 

Academic Editor

PLOS ONE